# Combined Recommendation Algorithm Based on Improved Similarity and Forgetting Curve

**Taoying Li *** , **Linlin Jin, Zebin Wu and Yan Chen**

School of Maritime Economics and Management, Dalian Maritime University, Dalian 116026, China;
jinlinlin@dlmu.edu.cn (L.J.); iwzebin@126.com (Z.W.); chenyan@dlmu.edu.cn (Y.C.)
***** Correspondence: litaoying@dlmu.edu.cn

**Abstract:** The recommendation algorithm in e-commerce systems is faced with the problem of high sparsity of users' score data and interest's shift, which greatly affects the performance of recommendation. Hence, a combined recommendation algorithm based on improved similarity and forgetting curve is proposed. Firstly, the Pearson similarity is improved by a wide range of weighted factors to enhance the quality of Pearson similarity for high sparse data. Secondly, the Ebbinghaus forgetting curve is introduced to track a user's interest shift. User score is weighted according to the residual memory of forgetting function. Users' interest changing with time is tracked by scoring, which increases both accuracy of recommendation algorithm and users' satisfaction. The two algorithms are then combined together. Finally, the MovieLens dataset is employed to evaluate different algorithms and results show that the proposed algorithm decreases mean absolute error (MAE) by 12.2%, average coverage 1.41%, and increases average precision by 10.52%, respectively.

**Keywords:** forgetting curve; combined recommendation; collaborative filter; similarity degree

## 1. Introduction

Recommendation system can be employed to provide users with personalized services, commodities or information, guiding users to browse targeted online goods or information, which makes the Internet from the past "people looking for information" to "information looking for people" intelligent stage [1]. With the gradual development of Internet to be more intelligent, recommendation system will be more and more widely applied in various fields, such as social network [2], information retrieval, text mining, personalization recommendation, biomedicine, and so on [3]. The rapid development of recommendation system in the e-business has also led to the academic research progress. Its main task is to improve the performance of recommendation algorithm and user satisfaction [3]. With the continuous development of recommendation systems, it is faced with many problems and challenges, such as the impact of negative evaluation on recommendation [4], emotional tendency of comments [5], trust evaluation of merchants [6] etc. However, data sparsity and user interest migration are the core problems needing to be solved. There are many studies on data sparsity, for example, Li, Deng et al. [7,8] partition users according to situational information for reducing the data dimension and sparsity of user scores. Ren, Qian, Lian et al. [9–11] proposed a joint probability model to simulate the decision-making process of user check-in behavior in order to deal with data sparsity. There are some recommendation algorithms considering the migration of user interests. For example, Zhu et al. [12] established a user interest model based on the non-linear progressive forgetting function in order to predict products' scores. Considering both data sparsity and user interest migration will be beneficial to improve the performance of personalization recommendations. The literature [13,14] listed a number of combinatorial recommendation methods. Yu and Li [15] defined the user's interests as short-term interest and long-term interest, and defined the weight

function based on the time-window as well, provide significantly better quality advice. Yin et al. [16] solved the problem of time effect quantification by quantifying the time effect of historical information in the recommendation system. These papers provided a good research direction.

Therefore, a combined recommendation algorithm based on improved similarity and forgetting curve is proposed for improving the accuracy of the results. Finally, the Movielens dataset was employed to evaluate the proposed algorithm by four indicators, including mean absolute error (MAE), mean square root error (RMSE), precision rate (Precision) and recall rate (Recall).

## 2. Materials and Methods

### 2.1. Collaborative Filtering Algorithm Based on Pearson

Collaborative filtering recommendation employs the behavioral data of other users, whose data is most similar to the target user, to predict how much the target user likes the item, which was used to determine whether the item is recommended to the target user [17]. This idea is consistent with people's behavior habits and it is easier for people to accept and recognize them [18].

Figure 1 summarizes and displays the general recommendation process of the collaborative filtering algorithm.

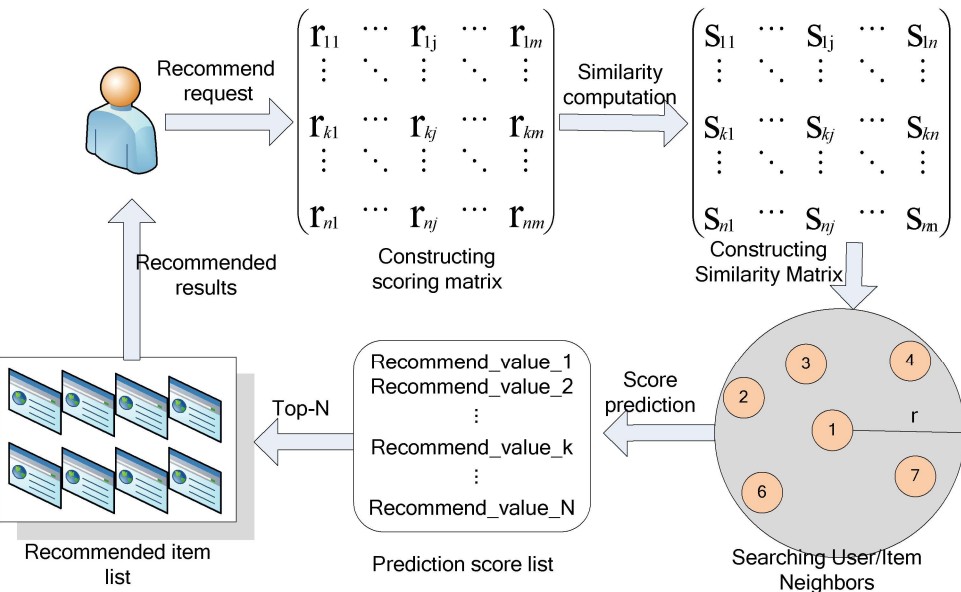

**Figure 1.** General recommendation process of collaborative filtering algorithm.

Collaborative Filtering Recommendation predicts the preferences of the target user based on the relevance of the items or the behavior of other users with same interests and experience. In other words, the traditional recommendation is an improved user collaborative filtering base on similarly (IUCF). Hence, collaborative filtering can be divided into user-based collaborative filtering (User-based CF) and item-based collaborative filtering (Item-based CF) [19,20].

For the user-based collaborative filtering, the larger the value of similarity between two users as showing Equation (1). If the similarity between two users is 1, it shows that they are completely similar and their habits are basically the same. Therefore, this paper focuses on improving user-based collaborative filtering recommendation. Since Pearson correlation coefficient is calculated based on the user's common scoring item. Therefore, when the score matrix is relatively dense, the number of items intersected by users is large and the reliability of the correlation coefficient is relatively high. On the contrary, the similarity calculated by users does not accurately reflect the degree of association between users. For example, the number of items intersected by users is very small, but the users' scores on these items are very similar. If a Pearson correlation coefficient is used to calculate, the degree

of correlation between two users is very high. If cosine similarity is used to calculate, the degree of similarity is relatively small. The calculated user similarity is too large or small, which cannot effectively reflect the actual degree of association between two users.

$$sim(u,v) = \frac{\sum_{i \in I_{u,v}} (R_{u,i} - \overline{R_u})(R_{v,i} - \overline{R_v})}{\sqrt{\sum_{i \in I_{u,v}} (R_{u,i} - \overline{R_u})^2} \cdot \sqrt{\sum_{i \in I_{u,v}} (R_{v,i} - \overline{R_v})^2}} \tag{1}$$

where, $R_{u,i}$ is user $u$'s score on item $i$, $I_{u,v}$ is a collection of common items between user $u$ and user $v$, $\overline{R_u}$ is the average of a user's score history. In the recommendation system, the advantage of using Pearson correlation coefficient calculation is that the similarity calculation is not affected when the user scoring standard is different, and the number of scoring items is not the same.

Therefore, when calculating the similarity among users, we should also consider the effect of the number of items with a common score on similarity. In general, the larger the proportion of items intersected by sets to all items scored by two users, the more similar these two users are to some extent.

In view of this, for user-based recommendation system, in order to prevent the influence of some behavior-intensive and hobby-wide users on Pearson correlation coefficient calculation under the condition of high data sparseness, a user-based collaborative filtering (UCF) algorithm based on Pearson is proposed by adding a user-related degree of weight factor to improve Pearson correlation coefficient. The weight is calculated by the proportion of a common score item between two users in its historical score items, and then it can calculate Pearson correlation coefficient by weighting for improving the accuracy of user similarity. Through this improvement, it can effectively reduce the impact of users who have scored many items on similarity, and to some extent alleviate the problem of data sparsity, so that the end-user neighbors and target users are closer to each other. In this paper, the improved Pearson correlation coefficient formula is shown as follows.

$$Sim(u,v) = \mu \cdot sim(u,v). \tag{2}$$

Formula (2) consists of two parts, the first is the weight of the user correlation degree $\mu = \frac{I(u) \cap I(v)}{I(u) \cup I(v)}$, the larger weight indicates that the improved Pearson correlation coefficient can reflect the degree of correlation between two users. The second is $sim(u,v)$, which is calculated according to the Pearson correlation coefficient of the Formula (1). According to the calculation formula, we know that $\mu \in (0,1]$; $I(u)$ represents a collection of items that have been rated by user $u$, and $I(v)$ represents a collection of items that have been rated by user $v$.

### 2.2. Recommendation Algorithm Based on Improved Ebbinghaus Forgetting Curve

The original recommendation system does not pay attention to the time, and the number of users is relatively small, moreover, the recommendation system is insensitive to the user's interest change under the influence of time. However, in recent years, with the development of the Internet, more and more attention has been paid to the interest mining of users, and the time-related factors have gradually begun to attract people's attention. Therefore, adding the influence of time on user interest into a recommendation system has become an important research topic in recommendation field [21].

In recent years, scholars have introduced time and user interest change factors into the relevant recommendation algorithms. Ding put forward arguments in his paper and presented that the overall interest of the user in the future was mainly influenced by his recent interests, so the recommendation algorithm should reduce the impact of early interest and increase the impact of the user's recent behavior on the recommended results [22]. Töscher proposed a dynamic ItemKNN algorithm, in which the nearest neighbor model in clustering was used to reflect the influence of different weight values of user interest [23]. Lu stated that the scoring vectors of users and items were regarded as a dynamic eigenvectors that change over time, and then the time factor was embedded in the recommendation model as a new one-dimensional to improve the performance of the recommendation system by

modifying the matrix decomposition model in the traditional collaborative filtering algorithm [24]. Lee et al. proposed a time-based collaborative recommendation system (TRS); TRS set a pseudo-score in the course of its work based on the shelf time of a commodity item and its time of purchase, and filled the score matrix with a pseudo-score to increase the density of the scoring matrix. The score filling mechanism of TRS reflected the effect of time on the recommendation system to a certain extent [25], but the TRS algorithm filled the score completely according to the predefined empirical scoring pattern, which required the prior knowledge of domain experts, so it was difficult to popularize and apply it on a large scale. Reference [15] and reference [16] had proposed the recommendation algorithm by combining the similarity and users' interest shift, which had shown that this idea is feasible. Hence, we try to use this idea to build a combination recommendation method.

### 2.2.1. Recommendation Considering Time

According to the analysis mentioned above, when calculating similarity, the shift of user interest with time will be considered. The time decay function is defined as Formula (2).

$$T(t) = (1 - \theta) + \theta \cdot \left( \frac{t - t_{\min}}{t_{\max} - t_{\min}} \right)^2 \tag{3}$$

where $t$ is the actual time for the user to score the item, and $t_{\min} \leq t \leq t_{\max}$, where $t_{\min}$ is the first time when the user evaluates in a recommendation system. Coefficient $\theta$ reflects a change in $T(t)$ interest, the greater the value of $\theta$, the faster the user's interest changes, the slower the reversal speed. When $\theta = 1$, the user's interest changes the most. When $0 < \theta < 1$, the change of interest tends to show an average trend. When $\theta = 0$, the user's interest remains unchanged. Therefore, the value of the $\theta$ corresponds to the speed of change of a user's interest in the recommendation system, which is positive proportional correlation.

Then, we will introduce the Ebbinghaus forgetting curve to improve the recommendation algorithm. Inspired by academic research and the Ebbinghaus forgetting curve, we can obtain that there are similarities between the changes in users' interest with time and the law of Ebbinghaus forgetting. Therefore, the Ebbinghaus forgetting function is used to simulate the users' interest change, and the user's score is calculated for interest migration to produce recommendations more in line with users' preferences. The traditional Pearson correlation coefficient only measures the linear correlation of two items in score, but does not distinguish the importance of score time characteristics in describing a user's interest [26]. This paper focuses on this aspect accordingly.

### 2.2.2. Correction of Ebbinghaus Forgetting Curve

The Ebbinghaus forgetting curve is shown in Figure 2.

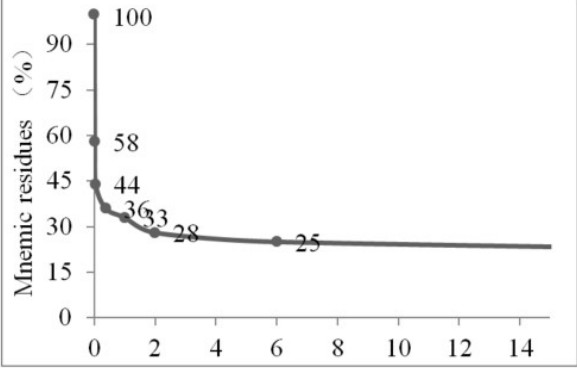

**Figure 2.** The Ebbinghaus forgetting curve.

From Figure 2, many people's forgetting of new memories is regular, which is not a uniform linear decreasing process, but in the initial stage, the forgetting speed is very fast, and then slows down gradually, at a certain time, the memory of things is kept at a relatively stable level, which is the law of memory forgetting. Many memory learning methods improve and utilize the law of forgetting curve, so as to achieve the effect of improving memory ability.

This pattern of oblivion is similar to the change in user interest, and our interest in an item can only last for a period of time, after that interest will be significantly weaker but not gone. Therefore, the curve of a user's interest changing with time can be simulated by the curve of Ebbinghaus forgetting law.

The Ebbinghaus forgetting curve [27,28]. can be shown in Formula (4).

$$f(x) = \frac{100k}{(\log x)^c + k} \tag{4}$$

where $x$ represents the oblivion time of the new memory, which is calculated by subtracting the score time of the specific item through the user's most recent score time. In order to obtain the value of two constants, $c$ and $k$, Ebbinghaus tested the amount of memory left by letting people remember given words. After repeated experiments, he concluded that the amount of memory left was a logarithmic function curve and $k = 1.84$, $c = 1.25$ [28]. Then, the formula and historical score data are used to construct a new evaluation matrix to complete the recommendation. In the subsequent experiment of this paper, the formula is compared with the improved time calculation formula to verify the validity of the improved formula.

In Figure 2, longitudinal coordinates are relative memory, while the horizontal coordinate is the time when memory begins to pass. Curves indicate a process in which people's relative memory storage gradually decreases over time. It can be seen that the law of meaningless letter forgetting is basically consistent with the basic time function Formula (4), both of which are non-incremental functions. So, the forgetting function is introduced to simulate the change of user's interest over time. The slower the forgetting function decreases, the greater the impact of the user's old interests on the recommendation results is, and the smaller the reverse. Therefore, the choice of forgetting function is very important, and will have a direct impact on the performance and effect of the recommendation system.

In order to obtain the best recommendation results, it should be consistent with the law of memory forgetting function in psychology while choosing the fitting function of forgetting law. Only by choosing a suitable function and applying it to the personalized recommendation system, can we provide users with more accurate recommendation results. Therefore, the forgetting curve fitting experiment will be carried out in this paper. The experiment is carried out under the environment of Matlab. The curve fitting toolbox cftool is used to fit the Ebbinghaus forgetting curve. The data in Table 1 is input as the basic data.

**Table 1.** Ebbinghaus forgetting experimental data.

| No | Time (Day) | Mnemic Residues (%) |
|----|-----------|---------------------|
| 1 | 0 | 100 |
| 2 | 0.0139 | 58 |
| 3 | 0.0417 | 44 |
| 4 | 0.375 | 36 |
| 5 | 1 | 33 |
| 6 | 2 | 28 |
| 7 | 6 | 25 |
| 8 | 31 | 21 |

At present, the main functions of simulating the user's interest forgetting process in some papers are exponential function and linear decreasing function. Referring to relevant academic papers,

different functions are used to adjust and fit the parameters. The fitting results of each function are shown in Figure 3.

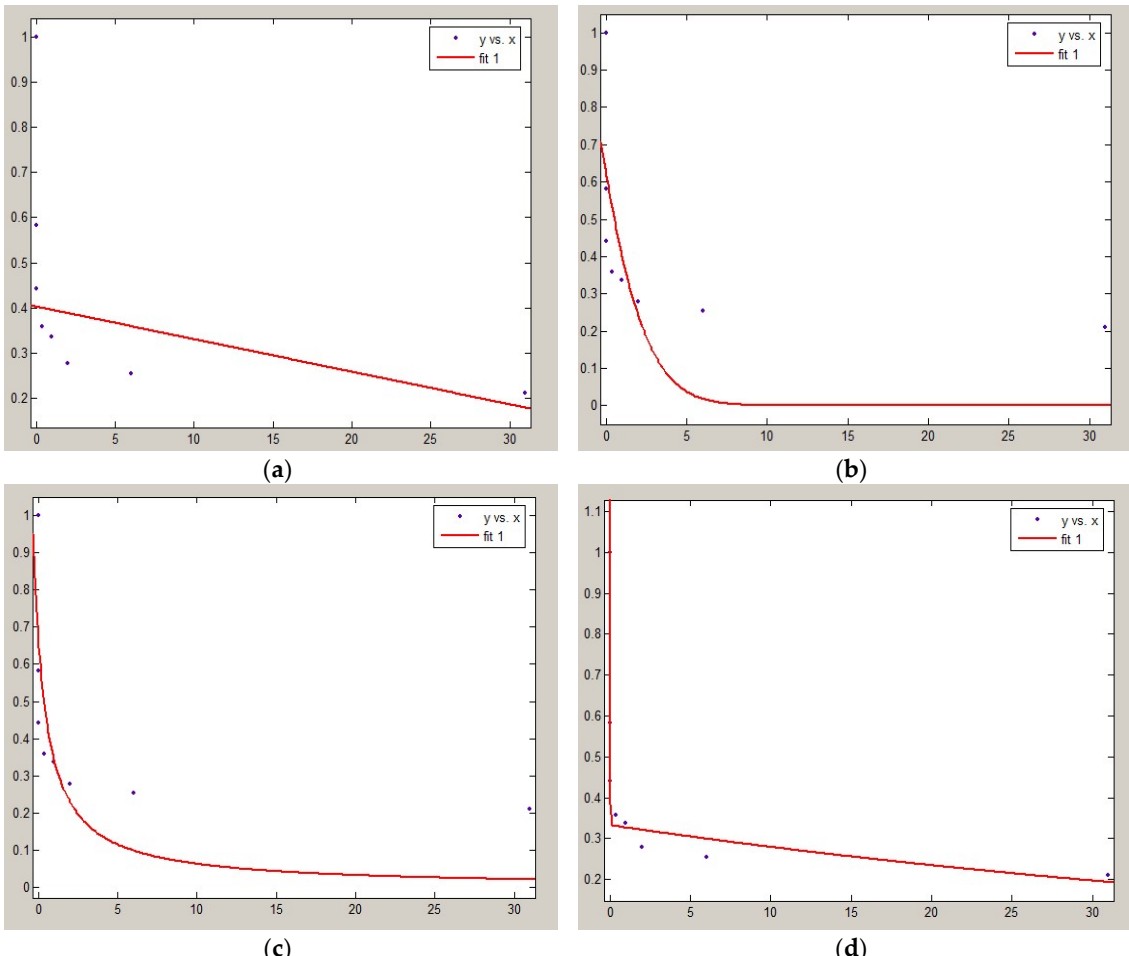

**Figure 3.** Fitting curves of different functions. They are listed as: (**a**) Linear function; (**b**) Gaussian function; (**c**) Rational function; (**d**) Exponential function.

Figure 3 shows that the shapes of linear function and Gaussian function are far from the Ebbinghaus forgetting curve, and they are obviously not suitable to simulate the forgetting law. The shapes of Rational functions and Gaussian function are close to the curve of forgetting law, but the rational function deviates further from the Ebbinghaus forgetting function after the point (0.375, 36%). Therefore, the image fitted by exponential function can accurately reflect the law of human forgetting.

After comparing the fitting effect of the image, according to the correlation coefficient of the fitting formula, the corresponding formulas of each function can be obtained as shown in Table 2.

**Table 2.** List of Fitting formulas.

| Functions | Formula |
|---|---|
| Linear function | $f(x) = -0.007206 \cdot x + 0.4027$ |
| Gaussian function | $f(x) = 2.75 \cdot e^{-\left(\frac{x+7.113}{5.843}\right)^2}$ |
| Rational function | $f(x) = \frac{0.7018}{x+1.047}$ |
| Exponential function | $f(x) = 0.6584 \cdot e^{-61.17 \cdot x} + 0.3325 \cdot e^{-0.01744 \cdot x}$ |

The experimental parameters of the fitting function are compared, and the final fitting formula is determined according to the fitness parameters. The main parameters include the sum of squares

of errors (SSE), root mean square error (RMSE), determination coefficient (R-square), and modified determination coefficient (Adjusted R-square). The smaller the values of SSE and RMSE (tending to 0 is the best), the smaller the error between fitting function and original function, the better the fitting effect; and the larger the values of R-square and Adjusted R-square parameters (tending to 1 is the best), the better the fitting effect of function. Fitness parameters of each formula after fitting function are shown in Table 3.

**Table 3.** List of parameters after fitting experimental.

|  | SSE | R-Square | Adjusted R-Square | RMSE |
|---|---|---|---|---|
| Multiform approximation | 0.2606 | 0.4383 | 0.3447 | 0.2084 |
| Gaussian approximation | 0.3079 | 0.3364 | 0.07102 | 0.2481 |
| Rational approximation | 0.2364 | 0.4905 | 0.4056 | 0.1985 |
| Exponential approximation | 0.01087 | 0.9766 | 0.959 | 0.05213 |

By comparing the image and the fitting suitability parameters of the Ebbinghaus forgetting curve, the results of SSE, R-square, modified coefficient and RMSE show that the fitting results of exponential function are more consistent with the forgetting curve than those of other functions. Therefore, the function of exponential function is chosen as the formula of Ebbinghaus forgetting curve.

$$f(x) = 0.6584 \cdot e^{-61.17x} + 0.3325 \cdot e^{-0.01744x} \tag{5}$$

where $f(x)$ is the retention rate of a memory after a broken time, and $x$ indicates the time elapsed from memory (Time unit: days).

### 2.2.3. Improved Recommendation Algorithm

The time user-based collaborative filtering (TUCF) proposed in this paper is based on the law of memory forgetting to track changes in user's interest. Human memory of new things is decreasing with the pass of time, so user's interest in a certain thing is in line with this trend, and the change of interest is reflected in the recommendation model by users' scoring data.

TUCF combines the time of user score and score data effectively to reflect changes in user's interest migration, so the processing of score time is a critical process. Therefore, for constructing the evaluation matrix, it is necessary to construct a corresponding time matrix as shown in Table 4.

**Table 4.** Score time matrix.

|  | User$_1$ | ... ... | User$_m$ |
|---|---|---|---|
| item$_1$ | T$_{11}$ | ... ... | T$_{1m}$ |
| ... ... | ... ... | T$_{ij}$ | ... ... |
| item$_n$ | T$_{n1}$ | ... ... | T$_{nm}$ |

In the Ebbinghaus forgetting curve, dependent variable $x$ is a period of forgetting time, while the time data in the score time matrix is the time node data. Therefore, before calculating the retention rate of user scores, a formula is introduced to process the user score time, and the user's active time in the system is processed accordingly, the formula is defined as Formula (6).

$$T(t) = \frac{T_{\max} - t}{T_{\max} - T_{\min}} \times T \tag{6}$$

where $T_{\max}$ refers to the user's most recent scoring time in the system, $T_{\min}$ refers to the user's earliest scoring time in the system, and $t$ is the current time to calculate the score ($T_{\min} < t < T_{\max}$). And $T$ is the time period (The unit is day), such as $T = 7$ means that the user's activity time is one week. Where the function $T(t)$ is monotonically non-increment, with the increase of the variable $t$, $T(t)$

becomes smaller, that is, the closer the current calculation time is from the user's most recent scoring time, the shorter t forgetting time experienced by the user, and the greater the amount of interest saved. Conversely, the farther away from the recent scoring time, the longer the forgetting time experienced, and the smaller the overall interest contribution to the user.

By combining the Ebbinghaus forgetting formula with the Formula (6), we can get the time influence function to calculate the user's interest retention rate based on scoring as Formula (7).

$$F(t) = 0.6584 \times e^{-61.17 \times \frac{T\max - t}{T\max - T\min} \times T} + 0.3225 \times e^{-0.01744 \times \frac{T\max - t}{T\max - T\min} \times T} \tag{7}$$

Considering that the time impact function described above is only an attenuation process for the user's interest, and does not take into account the weighting of the user's recent interest, the calculated forecast score will not conform to the true score of the user's recent interest. Therefore, this further improvement, in the above time formula to add an interest adjustment factor, the user's recent interest to adjust, so that the final forecast score more accurate. According to the Ebbinghaus forgetting curve, the user's memory retention rate in the first 2 days is large, after that the retention rate change tends to be flat, so set to 0–2 days between active intervals for user's interest. Therefore, a threshold is set; when the user's interest retention rate is greater than the threshold, the interest influence factor $\alpha$ is added to the time impact function, the reason for which is to increase the impact of recent interest, and then the score is weighted by the improved formula. When the user's interest retention rate for the item is less than the threshold, it is considered that the item does not match the user's current interest, but considers that the item user has scored, replaces the historical score with the mean will be more in line with the user's preference habits. The final formula is defined as Formula (8)

$$F(t) = 0.6584 \times e^{-61.17 \times \frac{T\max - t}{T\max - T\min} \times T} + 0.3225 \times e^{-0.01744 \times \frac{T\max - t}{T\max - T\min} \times T} + a \tag{8}$$

On the basis of Pearson collaborative filtering, combined with the law of the forgetting curve mentioned above, the change of user interest over time is considered when searching for target user neighbors and predicting score. The change of user interest in this paper is reflected in user scoring data. The time influence function based on Ebbinghaus forgetting is introduced into the Pearson coefficient, and the final improved user score similarity is defined as Formula (9), which expresses the correlation between users by their scores.

$$sim(u,v) = \frac{\sum_{i \in I_{u,v}} (R_{u,i} \cdot F(t_{u,i}) - \overline{R_u})(R_{v,i} \cdot F(t_{v,i}) - \overline{R_v})}{\sqrt{\sum_{i \in I_{u,v}} (R_{u,i} \cdot F(t_{u,i}) - \overline{R_u})^2} \cdot \sqrt{\sum_{i \in I_{u,v}} (R_{v,i} \cdot F(t_{v,i}) - \overline{R_v})^2}} \tag{9}$$

Eventually, the improved forecast formula is given as follows.

$$P_{u,i} = \overline{R_u} + \frac{\sum_{v \in N(u)} sim(u,v) \cdot (R_{v,i} \cdot F(t_{v,i}) - \overline{R_v})}{\sum_{v \in N(u)} |sim(u,v)|} \tag{10}$$

$I_{u,v}$ represents a collection of items that user $u$ and user $v$ have jointly scored, $t_{u,i}$ indicates the score time of user $u$ for item $i$, $\overline{R_u}$ represents the evaluation score of user $u$. $F(t_{u,i})$ calculates the amount of interest retention after the historical score of user $u$ is removed from the center, so that the calculated score is more in line with the user's current interests.

*2.3. Combined Recommendations of Improved Similarity and Forgetting Curve*

The improved Pearson similarity can effectively enhance the similarity calculation under the condition of data sparse, so the similarity degree after calculation is more accurate and the recommendation quality is greatly increased. The improvement based on time impact is a collaborative filtering algorithm, which can effectively simulate the migration of user's interest, and through the

processing of user scores, not only the recommended accuracy but also the coverage of the system have been greatly improved.

There are many references for improving the quality of recommendation by focusing on the change of interest with time, for example, Margaris and Vaz [29,30] considered the impact of the time for improving prediction quality by using datasets of movies, music, videogames and books, Chen and Lo [31,32] studied on user rating for tracking their interest shift, and Vinagre [33] used forgetting mechanisms for scalable collaborative filtering.

Therefore, in this section, the methods of the collaborative filtering algorithm and Ebbinghaus forgetting curve are combined according to the various methods of mixed weighting, which can reflect the user's interest. Therefore, the combined recommendation based on time and improved similarity is proposed.

The recommendation algorithm based on similarity and time influence can effectively improve the accuracy of recommendation and reduce the error of prediction score. It shows that the effect of reducing the number of users with too many scores, and adding time factors can effectively increase the recommended quality. After a number of fusion methods, two algorithms are combined by weighting, stacking and so on, mainly from two aspects of similarity calculation and prediction score.

(1) The similarity calculation is enhanced by two methods respectively, and the recommendation results based on the improved similarity of the user are more accurate than the effect based on the time effect. Therefore, under the condition of sparse data, the improved similarity can more accurately match the correlation among users, so it is more appropriate to use improved user similarity in similarity calculations.

(2) For predicting user scores, on the basis of traditional similarity, the historical score data of the nearest neighbor of the target user is processed based on time influence, so the score is more in correspondence with the interest of the user's score, and effectively enhances the recommendation effect.

Finally, the similarity coefficient calculation of the fusion algorithm is based on the similarity calculation of the frequent user penalty. If the user is rated, the user's historical score is calculated according to the time influence function, and the final scoring forecast list is obtained.

## 3. Results and Discussion

The 80,000 records chosen from the 100K dataset provided by Movielens are selected as training data, and the remaining 20,000 records are selected as test data.

The experiment in this paper is divided into three aspects.

### 3.1. Resutls of the Improved Similarity Recommendation Algorithm

In the experiment, we set the length of recommendation list $N$ maximum to 100, if the recommendation list is too long, it is impossible for users to browse it all. For obtaining MAE and RMSE, the number of user's neighbors is variable from 5 to 300, and the initial interval is 5, the interval after 50 is set to 10 and after the number of neighbors 100, the interval is set to 100. The comparison results of MAE and RMSE are shown in Figures 4 and 5.

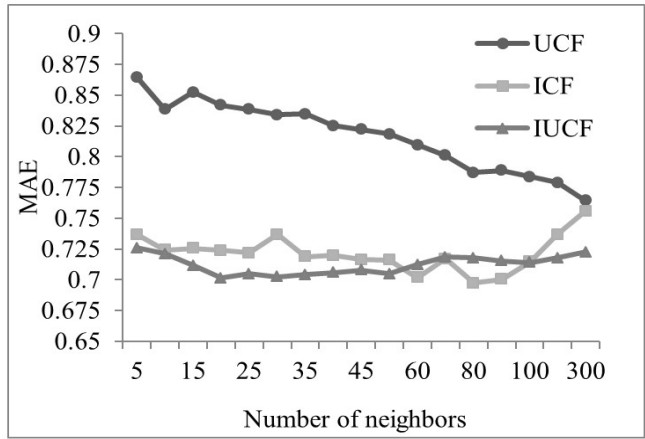

**Figure 4.** Comparison of the changes in MAE with the number of neighbors.

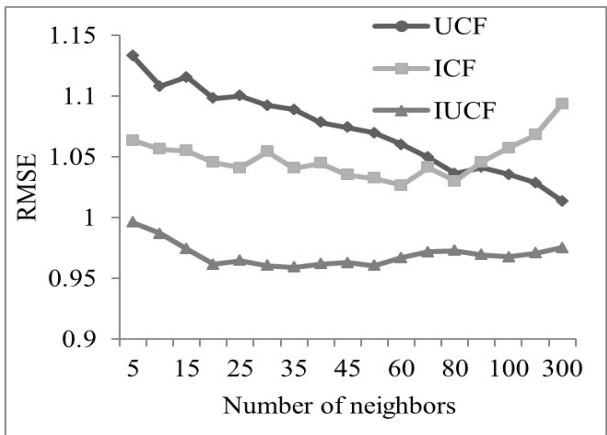

**Figure 5.** Comparison of changes in RASE with the number of neighbors.

In Figures 4 and 5, the algorithm of improved user-based collaborative filtering (IUCF) has enhanced greatly in the MAE index than the traditional user-based collaborative filtering (UCF). Compared to the item-based collaborative filtering (ICF), it also has a good performance, only in the number of neighbors in 60~90 slightly worse than ICF. For RMSE as shown in Table 5, the IUCF has lower errors then UCF and ICF, which shows that the improved algorithm has better performance in the environment with high scoring requirements.

**Table 5.** Statistical results of RMSE.

|  | Maximum Value | Minimum Value | Average Value |
|---|---|---|---|
| UCF | 1.13306 | 1.0134 | 1.0719 |
| ICF | 1.0936 | 1.0265 | 1.0488 |
| IUCF | 0.9959 | 0.9589 | 0.9695 |

For Precision and Recall, based on the above experiments, combined with the performance requirements of the MAE and RMSE indicators, the nearest neighbor number for UCF is 20 and those for ICF and IUCF are 25 and 20, respectively. The reason is that the corresponding algorithm has good recommendation accuracy when the number of neighbors is taken as these numbers. Then, according to the experiment need Top-*N* to select the integer among 5–100, the front 50 size when the recommendation list length *N* interval is 5, the interval after 50 is set to 10. Because of the contradiction between the Precision and the Recall indicators (shown in Figures 6 and 7); finally, the experiment in Figure 8 is compared with the F1 indicator that is the formula with the $\alpha$ value of 1 in the F-Measure indicator.

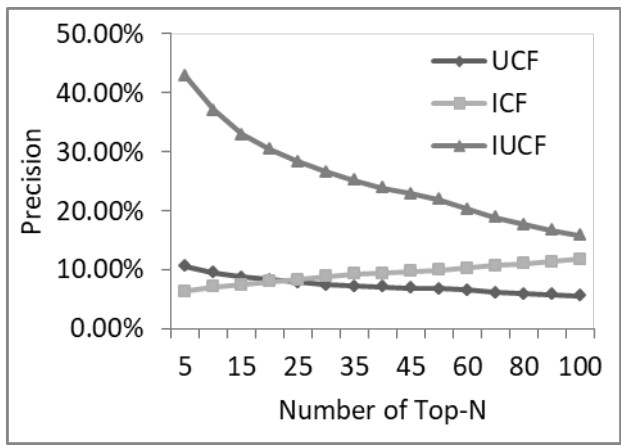

**Figure 6.** Comparison of precision with Top-*N* changes.

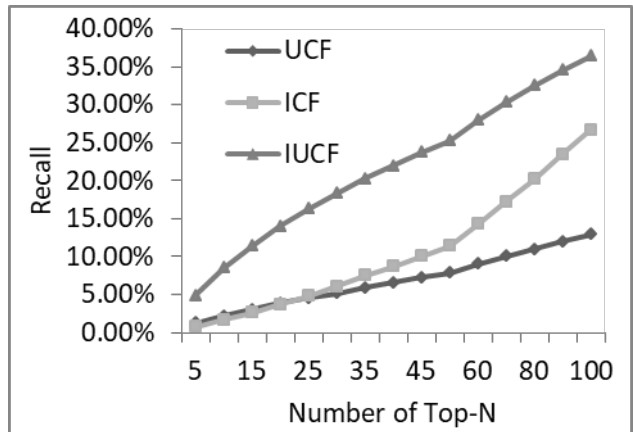

**Figure 7.** Comparison of recall with Top-*N* changes.

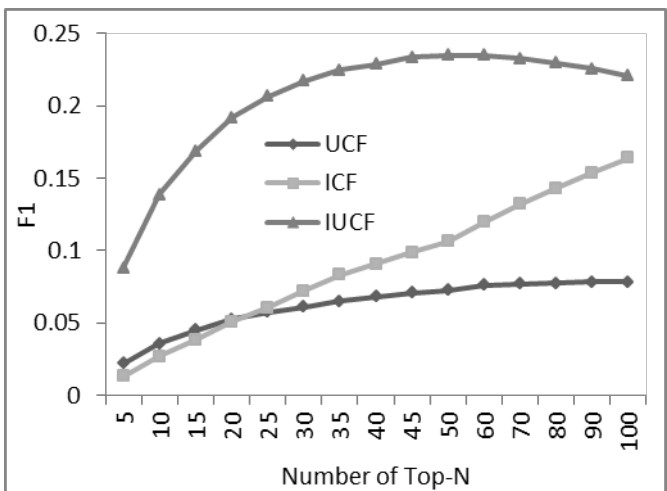

**Figure 8.** Comparison of F1 with Top-*N* changes.

Through the above experimental results in Figures 6–8, it can be seen that in the Precision and Recall indicators, the improved algorithm in this paper is obviously better than the two, indicating that in the case of the same recommendation list length, the improved algorithm can more accurately hit the user needs and can have a more comprehensive coverage of the user. In the F1 index, the improved algorithm IUCF has a more stable performance than UCF and ICF in the case of weighing Precision and Recall.

From the verification of the above experiments, the improvement of this paper effectively improves the progress of the recommended forecast, hits and covers the user interest preference of the items.

### 3.2. Results of the Improved Ebbinghaus Forgetting Curves

The training data and test data are also divided like the above section, and the experiment is evaluated according to the corresponding indexes introduced in the second section. In order to verify the influence of threshold value and interest weighting factor on time-based algorithm, the threshold value is obtained from Ebbinghaus forgetting data under the condition of recommended list length $N = 25$ and the user's nearest neighbor number $K = 20$. The threshold is obtained from the Ebbinghaus forgetting data. The improved score is tested among 0.2~1.1 for the interest weighted actor $\alpha$. The experimental results are obtained as follows.

From Figures 9 and 10, the threshold value is set to 0.33, with the change of interest weighted factor value, and the prediction error of the algorithm is low while the accuracy is high. By observing the change curves of MAE and Precision, we know that if $\alpha$ is close to 0.8, the error value of the algorithm is the smallest and the accuracy is the highest, so the recommendation algorithm based on time impact will have better recommendation performance. In the following comparison experiment, the time-based algorithm is taken under the optimal coefficient of the above experiment.

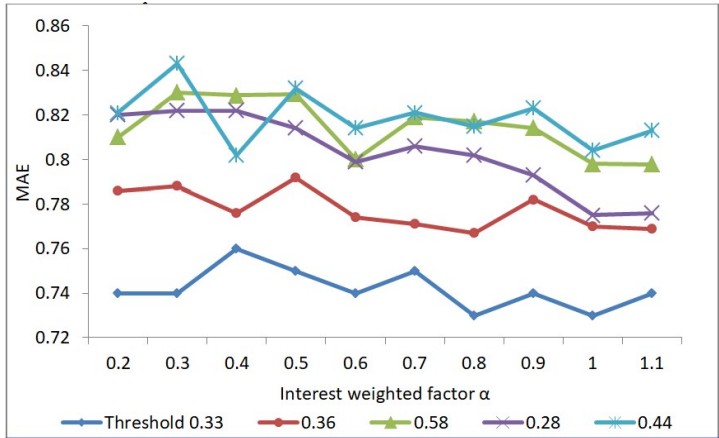

**Figure 9.** MAE with threshold and interest weighted factor.

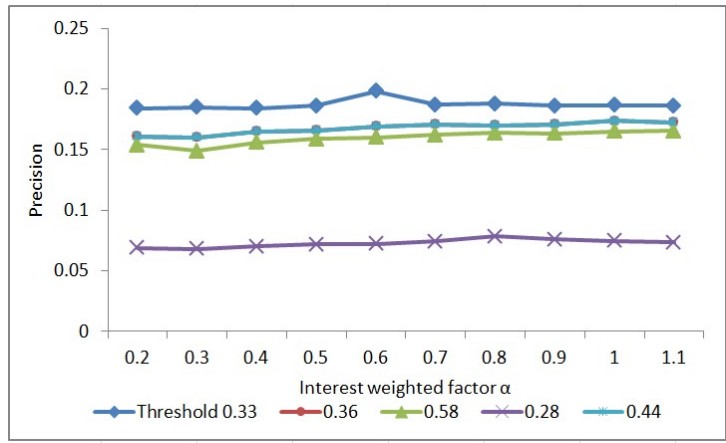

**Figure 10.** Precision with thresholds and interest weighted factors.

Comparative experiments are carried out on user-based collaborative filtering (UCF), recommendation based on a traditional Ebbinghaus forgetting formula (TCF1), and recommendation

based on improved formulas (TCF2). The recommendation list length *N* is 100, the number of user neighbors is chosen from 5 to ~300, the experimental results are given in Figures 11 and 12.

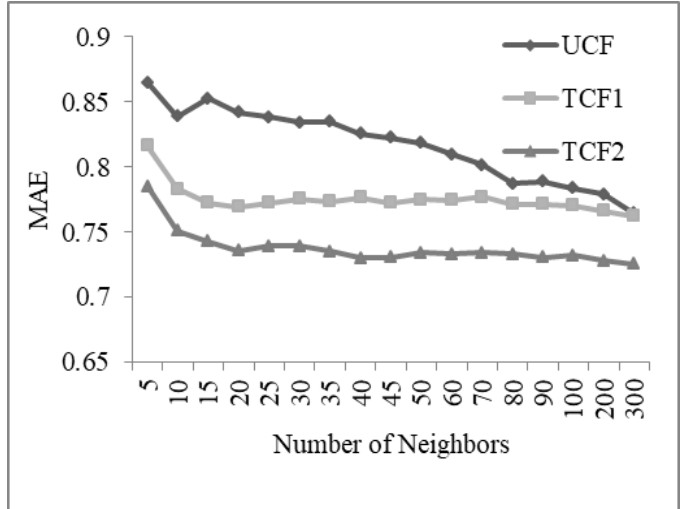

**Figure 11.** MAE with changing of the number of neighbors.

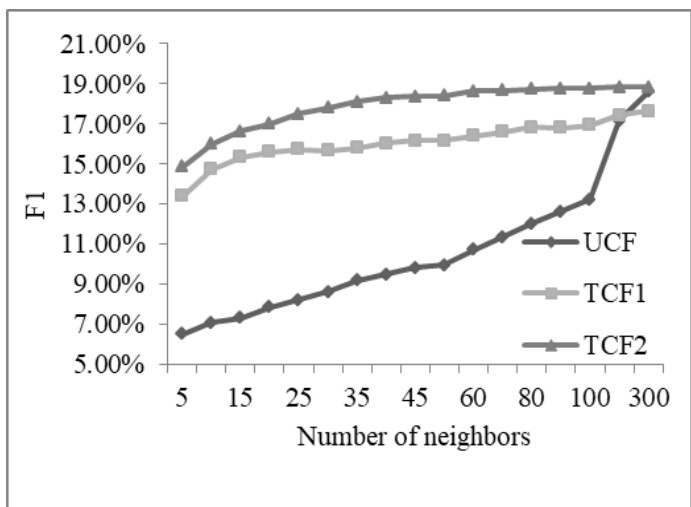

**Figure 12.** F1 with changing of the number of user neighbors.

For MAE, the forgetting curve is employed to improve the processing of the historical score for increasing the accuracy of the recommendation system. The time-based collaborative filtering recommendation (TCF2) has a lower error than the recommendation based on the traditional Ebbinghaus forgetting formula. It allows for more accurate prediction of user scores, indicating that improvements in this paper are effective.

For F1, the proposed method has a high F1 value compared with UCF and TCF1, which shows that the proposed algorithm based on time effects has better recommendation performance and can cover user preferences more accurately when considering both accuracy and recall. Other specific statistical parameters of the experiment are shown in Table 6.

**Table 6.** Statistical results on performance of different methods.

|       | Min MAE | Avg MAE | Max Precision | Avg Precision | Max Recall | Avg Recall |
|-------|---------|---------|---------------|---------------|------------|------------|
| UCF   | 0.7642  | 0.8167  | 0.1335        | 0.0759        | 0.3064     | 0.1734     |
| TCF1  | 0.7694  | 0.7751  | 0.1266        | 0.1154        | 0.2905     | 0.2647     |
| TCF2  | 0.7254  | 0.7376  | 0.1353        | 0.1285        | 0.3104     | 0.2949     |

*3.3. Results of the Combined Recommendation Algorithm*

Hybrid/combined improved time and user-based collaborative filtering (HITUCF) is the recommendation method based on both forgetting curve and similarity. The data in this section is randomly selected from Movielens, but it needs to be met that each user has scored at least 20 movies, with a total data of 100,000 records. In order to test the stability of the algorithm in different data, a random division of the dataset is taken five times for five-fold cross-validation. The tests for each partition are different from the other test set data, dividing the data by choosing 80% as the training data and 20% as the test data. In order to eliminate the effect of other factors on the experiment, the number of neighbor sets is set to 20, and the length of recommendation item list for users Top-*N* set to 25. The experimental results are compared by MAE, precision and coverage indicators, shown in Figures 13–15.

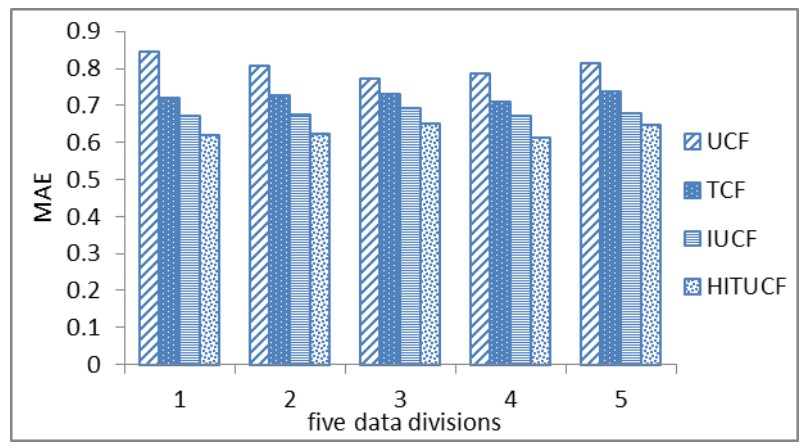

**Figure 13.** MAE of five data divisions.

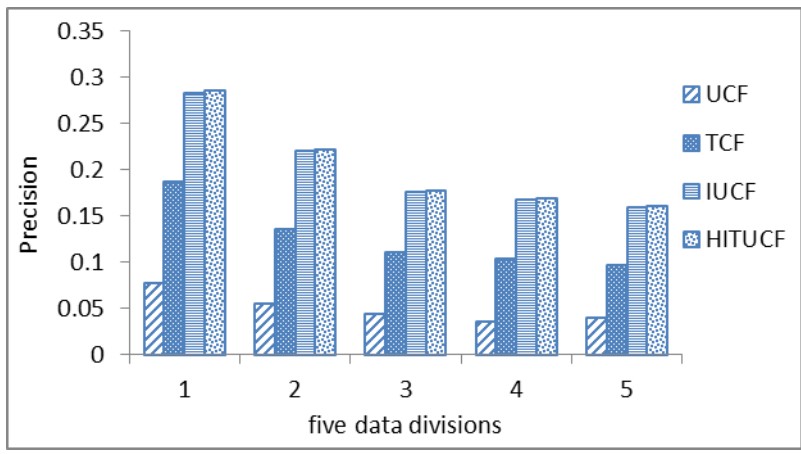

**Figure 14.** Precision of five data divisions.

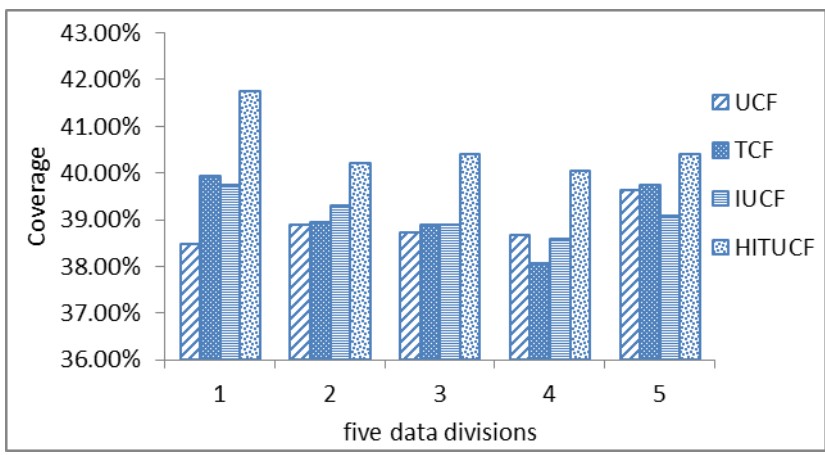

**Figure 15.** Coverage of five data divisions.

From Figures 13–15, according to five experiments on randomly grouping data, it is found that the improved algorithm proposed in this paper has lower error and higher accuracy than the traditional collaborative filtering algorithm, and in the coverage aspect, it also has good effects, which means that the proposed algorithm is relatively stable and has the good performance, and it is more effective for mining long tail items.

According to the change of the user's neighbor set, a comparative experiment is carried out on the implemented algorithm, where $N = 25$, the number of neighbors is to take 5~100, and the recommendation results are shown in Figure 16.

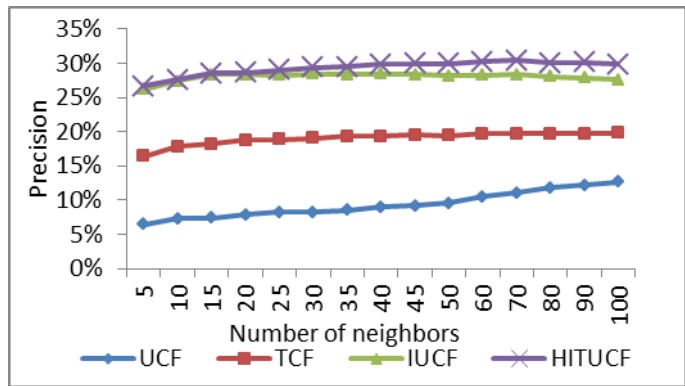

**Figure 16.** Relation between precision and number of neighbors on 100K Movielens dataset.

In order to show the performance of the proposed methods, the 1M Movielens dataset is then used to do the experiment again. The number of neighbors is to take 10~100, and the recommendation results are shown in Figure 17.

From Figures 16 and 17, we know that the precision of recommendation results goes up with the increase of the number of neighbors. The improved algorithm is obviously superior to other algorithms, which shows that the improved algorithm has a significant improvement.

The HITUCF proposed in this paper are more accurate than the traditional collaborative filtering recommendation, and can effectively increase the recommendation quality, meanwhile it reduces the average error.

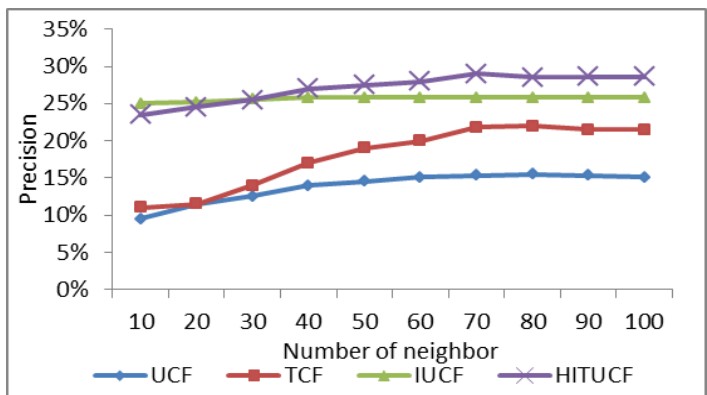

**Figure 17.** Relation between precision and number of neighbors on 1M Movielens dataset.

## 4. Conclusions

User interest migration and data sparsity are crucial problems to be solved in the recommendation algorithm. Hence, in this paper, a combined recommendation algorithm based on improved similarity and forgetting curve is proposed. Firstly, it corrects the similarity of Pearson under the condition of sparse data. Secondly, the user scoring time is introduced into the recommendation system as an important factor, and the recommendation algorithm based on an improved Ebbinghaus forgetting curve is established; Finally, the Movielens dataset is used to verify the algorithms, and the combined recommendation algorithm has shown better results by comparing with UCF, TCF, IUCF algorithm.

Online shopping has become popular in recent years. Nowadays, a product introduction page of shopping software is opened randomly. The "Guess what you like" section will show dozens of products, some of which are the same products of different stores, and some of them are completely different substitutes, which are supported by the recommendation algorithm to provide customers with more product choices. Therefore, how to deal with the change of user interest with time, recommending favorite goods or providing information to users becomes the focus of e-commerce.

**Author Contributions:** Conceptualization, T.L.; methodology, L.J.; software, L.J.; validation, Y.C.; writing—original draft preparation, T.L.; writing—review and editing, L.J. and Z.W.; project administration, T.L and Y.C.; funding acquisition, T.L and Y.C.

**Funding:** This research was funded by the National Natural Science Foundation of China, grant number 71271034, the National Social Science Foundation of China, grant number 15CGL031, the Fundamental Research Funds for the Central Universities, grant number 3132016306 and 3132018160, the Program for Dalian High Level Talent Innovation Support, grant number 2015R063, the National Natural Science Foundation of Liaoning Province, grant number 20180550307, and the National Scholarship Fund of China for Studying Abroad.

**Conflicts of Interest:** The authors declare no conflict of interest.

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
