# Peer review of "Combined Recommendation Algorithm Based on Improved Similarity and Forgetting Curve"

_information, doi:10.3390/info10040130_

Reviewer 1 Report

The legend in figure should be neat and rearranged. Ex. Figure 7, 9,10

Research design and time-based embedded forgetting are well done. Formula of each Model fitting function, and residual analysis compared are clearly discussed.

Please add some academic contribution and practical application based on the results in the conclusion.

Author Response

The legend in figure should be neat and rearranged. Ex. Figure 7, 9,10

Response: we have revised it.

Research design and time-based embedded forgetting are well done. Formula of each Model fitting function, and residual analysis compared are clearly discussed.

Please add some academic contribution and practical application based on the results in the conclusion.

Response: we have revised it.

Reviewer 2 Report

It is a novel idea to add the Ebinhaus forgetting curve to similarity calculation of CF recommendation, which will represent the timeliness of user scores more scientifically. However, there are still some possible improvements in the paper. 

In Section 2.2.2, 4 different formulas are tested to determine the best to fit the Ebbinghaus forgetting curve. As the Ebbinghaus forgetting curve seems regular on both the falling interval and the flat interval, respectively, a piecewise function may achieve better effect.

In Section 2.2.3, a factor α is finally added to the formula of Ebbinghaus forgetting curve F(t) . The reason for using parameter Î± may not have been explained clearly yet. It is suggested that some explanation ways other than text description be used for introducing this parameter.

But on the whole, this paper has bought a new way to deal with data sparseness and timeliness issues. The result shows that the presented method is of high stability and acts better on those users with less data. This paper may be useful to research on user dynamic interest.

Author Response

It is a novel idea to add the Ebinhaus forgetting curve to similarity calculation of CF recommendation, which will represent the timeliness of user scores more scientifically. However, there are still some possible improvements in the paper.

In Section 2.2.2, 4 different formulas are tested to determine the best to fit the Ebbinghaus forgetting curve. As the Ebbinghaus forgetting curve seems regular on both the falling interval and the flat interval, respectively, a piecewise function may achieve better effect.

Response: Thanks to the reviewers. We will consider it in future research.

In Section 2.2.3, a factor α is finally added to the formula of Ebbinghaus forgetting curve F(t) . The reason for using parameter α may not have been explained clearly yet. It is suggested that some explanation ways other than text description be used for introducing this parameter.

Response: the reason for factor α is to increase the impact of recent interest.

But on the whole, this paper has bought a new way to deal with data sparseness and timeliness issues. The result shows that the presented method is of high stability and acts better on those users with less data. This paper may be useful to research on user dynamic interest.

Response: Thanks to the reviewer’s comments.

Reviewer 3 Report

This paper proposes an improved method for callborative filtering based on the forgetting curve. The paper has an extensive experimental section wiith several valuable insights. However I do not feel that the paper is mature enough for publication at this stage for the following reasons:

- I am pretty sure that other contributions of recommender systems based on the forgetting curve have been made in the past, however none of them are cited or used for comparison.

- One of the main features (if not the main feature) of the forgetting curve model is to account for periodic reinforcement of memory. The authors dismiss this feature (lines 179-181) stating that this is not applicable to recommender systems, which is arguable to say the least. I can think of several applications where this could be useful (essentially any application where users interact multiple times with the same items).

- Experiments are performed using a single dataset (ML-100k), which is a severe limitation. Furthermore, this is a rather small dataset for today's standards.

- The parameters for the forgetting curve are based on [26], however there is no evidence whatsoever that these parameters are valid for any other study than [26].

- It is not possible to understand the difference between UCF, ICF and IUCF from the text.

- The parameter \mu in eq (2) is not well described. What are the meanings of I(u), I(v) and I(v)'?

- Legend for two of the bars are missing in Figs 13-15

 Author Response

This paper proposes an improved method for callborative filtering based on the forgetting curve. The paper has an extensive experimental section wiith several valuable insights. However I do not feel that the paper is mature enough for publication at this stage for the following reasons:

- I am pretty sure that other contributions of recommender systems based on the forgetting curve have been made in the past, however none of them are cited or used for comparison.

Response: We have added two references [7][8].

- One of the main features (if not the main feature) of the forgetting curve model is to account for periodic reinforcement of memory. The authors dismiss this feature (lines 179-181) stating that this is not applicable to recommender systems, which is arguable to say the least. I can think of several applications where this could be useful (essentially any application where users interact multiple times with the same items).

Response: We have removed that sentence.

- Experiments are performed using a single dataset (ML-100k), which is a severe limitation. Furthermore, this is a rather small dataset for today's standards.

Response: Thank you for the reviewer's suggestion. We will choose other larger data sets for future research.

- The parameters for the forgetting curve are based on [26], however there is no evidence whatsoever that these parameters are valid for any other study than [26].

Response: There are many references setting K = 1.84, C = 1.25, which we can show by querying "K = 1.84, C = 1.25" on Google Scholar. We only chose the most frequently cited literature in our paper.

- It is not possible to understand the difference between UCF, ICF and IUCF from the text.

Response: UCF is a recommendation method based on similarity among users. ICF is a recommendation method based on similarity among items. IUCF is a UCF method based on improved similarly by using Pearson correlation coefficient.

- The parameter \mu in eq (2) is not well described. What are the meanings of I(u), I(v) and I(v)'?

Response: I(u) represents a collection of items that have been rated by user u ,and I(v) represents a collection of items that have been rated by user v.

- Legend for two of the bars are missing in Figs 13-15

Response: we have added the missing legend from Fig 13-15.

Round  2

Reviewer 3 Report

The concerns I expressed in the first review have been partially addressed. The following problems persist (by order of importance):

- Experiments were conducted on a single - and small - dataset. This is the biggest limitation of the paper, however the authors have deliberately dismissed it in their rebuttal.

- The application of the forgetting curve to this work is admittedly partial. 

- References [7] and [8] were fortunately added, however it is not clear where this work is different from that previous work.

- Hyperparameter tuning by searching Google Scholar is not acceptable in journal papers, in my opinion.

- Language has not been significantly improved.

Given the small improvements of the manuscript, I have upgraded my recommendation to "Reconsider after major revision". 

Author Response

We have provided a point-by-point response to the reviewer’s comments.

Round  3

Reviewer 3 Report

In this new version the authors have addressed most of my concerns. I am changing my recommendation to acceptance with minor reviews, mostly because I detected a few minor problems in language, so I would advise a second language review.

I also recommend authors that in future submissions, especially to journals, a more extensive experimental study is provided, with a larger variety of modern datasets. Movielens is an outdated, small dataset (even the 1M version) under today's standards.